# Teaching and Learning Methods in Geography Promoting Sustainability

**Eija Yli-Panula [1],\*** , **Eila Jeronen [2]** **and Piia Lemmetty [1]**

1   Department of Teacher Education, University of Turku, 20014 Turku, Finland; piia.lemmetty@utu.fi
2   Faculty of Education, University of Oulu, 90014 Oulu, Finland; eila.jeronen@oulu.fi
\*   Correspondence: eija.yli-panula@utu.fi

**Abstract:**  Understanding and learning geographic knowledge and applying it to sustainable development (SD) depends not only on the knowledge itself, but also on how it is taught and studied.  The teaching and learning methods for promoting sustainability in geography have not been thoroughly studied.  This qualitative study examined articles on geography teaching and SD. The material was selected using keywords related to geography education. The study describes 17 articles published in peer-reviewed scientific journals from 2008 to 2018. The focus group varied from primary to higher education.  The proportion of teaching and learning methods were determined.  The data were analyzed using qualitative content analysis.  The foci of the analyses were the teaching and learning methods, topics, goals, and levels of thinking skills.  Additionally, features of the teaching methods used in geography education, including outdoor education, to achieve the sustainable development goals (SDGs) were investigated. Different teaching methods used together and interactive learning were the most often preferred.  Group work and teachers' presentations were mentioned in 12 articles, and inquiry-based learning and argumentation in half of the articles.  The most often written expressions promoting SD in geography education concerned environmental sustainability (42%), followed by social (25%), economic (19%), and cultural sustainability (14%).  The most emphasized features of the current teaching methods were active participation, thinking skills, animation, evaluation, dialog, demonstrations, and information and communication technology skills. The whole school approach and forward-looking perspective in geography should be implemented in school education to reach the SDGs and to support SD.

**Keywords:** general education; geography education; higher education; literature review; outdoor education; sustainability education

## 1. Introduction

The International Charter on Geographical Education defines geography as, "the study of Earth and its natural, physical, and human environment" [1]. Geography involves the study of human activities and their interrelationships and interactions with environments on local to global scales. It bridges the natural and social sciences, and thus deals with spatial variability, that is, that phenomena, events, and processes vary within and between places and therefore should be regarded as an essential part of the education of all citizens in all societies. The abovementioned bases of geography can be regarded as supporting sustainability education (SE). According to Day and Spronken-Smith [2], interdisciplinary approaches which integrate three aspects—the economic, social, and physical aspects of sustainability—are not only well suited for geography, but remain the pre-conditions for understanding its multiple dimensions.

The core concepts in geography education are space, place, landscape, and sustainability [3]. Furthermore, sustainability can be conceptualized as a core geographical concept and also as a cross-curricular theme. In SE, sustainable development (SD) exercises can generate holistic experiences around philosophical, theoretical, and practical sustainability issues. Examples of the latter are forest schools [4], learning outside the classroom [5], and a range of creative approaches to education [6].

Education is seen as a key strategy for achieving sustainable development. Global Education and SE can help students develop their critical thinking skills and values and help them to understand a globalized and interdependent world and their own rights and responsibilities towards each other and the planet. These are the topics in human geography which must be comprehended by learners if they are to reach these goals in geography. The environmental anxiety caused by environmental problems has increased the need for SE. For their part, decision-makers should promote citizens' commitment to the environment and 'ecological literacy' [7–9]. According to McBride, Brewer, Berkowitz, and Borrie [10], ecological literacy frameworks emphasize systems thinking, which involves identifying the various biophysical and social components in a given environmental context and distinguishing their interrelations. An ecologically literate individual has a clear perception and understanding of a system's dynamics and ruptures, as well as its past and alternate future trajectories. They understand the complexity of studied objects and phenomena, thus enabling better decision-making. Higher order thinking skills, such as systems thinking, can be developed, for example, by place-based education (PBE) [11].

According to Woodhouse and Knapp [12], PBE refers to community-focused schooling, ecological education, and bioregional education. Stone [13] argued that PBE is fundamental to schooling for sustainability and that sustainability should be holistic in its approach. He also argued that familiar and beloved places are most likely to be protected and preserved for future generations.

Teaching SD in geography depends not only on high-quality subject matter knowledge, but also on modern researched pedagogical content knowledge, which means teachers' interpretations and transformations of subject matter knowledge in the context of facilitating student learning [14]. However, the teaching and learning methods in connection to education for SD in geography have not been thoroughly studied. This study should fill this gap in the research. The aim of this study was to investigate the most supported teaching and learning methods which promote sustainability.

## 2. Theoretical Background

### 2.1. Geography and Sustainability Education

Many geographers are critical of the concepts of environmental education (EE), education for sustainable development (ESD), and sustainable development education (SDE) [15–17]. The definition of concepts varies depending on the context [18]. According to Lucas [19], EE is a lifelong learning process which aims to raise environmental awareness and to promote local, regional, and global environmental activities [20]. According to UNESCO [21], ESD aims to make people aware of current and future environmental problems and challenges, and to create more sustainable and resilient societies. SDE, for its part, looks at the interaction between ecological and social systems with a view to developing solutions to a variety of unexpected situations [22]. All three are focused on quality education and a society that takes into account the carrying capacity of the globe [23]. Thus, they can be considered to include all dimensions of SD. In this study we use the term sustainability education (SE) [24], as it can be thought to include all forms of EE, ESD, and SDE.

In her study, Pauw [25] described future societies as rapidly changing, interconnected, interdependent, competitive, individualized, and knowledge-intensive. Geography and sustainability education offer ways to understand this increasingly complex and unsafe world. Based on geographic information, it is possible to predict order, uncertainties, crisis, and chaos. Geography's tradition in the human environment theme provides a methodological basis for dealing with issues of sustainability and geographical approaches concerning dynamics, complexity, and interactions which support the

understanding of the spatio-temporal dimensions of sustainability. In addition, geography also bridges, by its interdisciplinary approaches, the social and natural sciences [15]. Education in geography fosters knowledge, skills, and concepts for better understanding our being, our relationships with other people, and the universe [26]. So, geographic awareness and geographic consciousness are also important goals in SE cf. [27].

## 2.2. Geographical Competencies and Skills for Sustainable Development

The International Charter on Geographical Education [1] states that studying geography should support young people to understand and appreciate how places and landscapes are formed, how people and environments interact, the consequences that arise from our everyday spatial decisions, and Earth's diverse and interconnected mosaic of cultures and societies. According to Haubrich [28], the achievement of SD requires geographic competences such as knowledge and understanding of the major natural systems of the Earth (landforms, soils, water bodies, climate, vegetation) and the interactions within and between ecosystems and the major socio-economic systems of the Earth (agriculture, settlement, transport, industry, trade, energy, population, and others).

Valuable geographical skills are the ability to communicate, think about, and use practical and social skills to explore geographical topics on a range of scales, from local to international. In addition, attitudes and values dedicated to seeking solutions to local, regional, national, and international problems on the basis of the Universal Declaration on Human Rights are indispensable [28,29]. Moreover, teaching about SD means teaching holistically. Geography, as a bridge between natural and human sciences, is used to practice such an approach. Geographical education contributes to this by ensuring that individuals become aware of the impact of their own behaviour and that of their societies, and that they have access to accurate information and skills to enable them to make environmentally sound decisions and to develop an environmental ethic to guide their actions [29].

## 2.3. Learning Topics and Interdisciplinary Skills for Sustainable Development through Geography Education

The Lucerne Declaration states that the themes of the UN Decade of Education for Sustainable Development (DESD) 2005–2014 have much in common with geography's objects of study [30]. The United Nations Educational, Scientific, and Cultural Organization (UNESCO) is involved in monitoring the progress towards Target 4.7 of SDG 4 on Education, which focuses on Global Citizenship Education (GCED) and Education for Sustainable Development (ESD). Target 4.7 states: "By 2030, ensure that all learners acquire the knowledge and skills needed to promote sustainable development, including, among others, through education for sustainable development and sustainable lifestyles, human rights, gender equality, promotion of a culture of peace and non-violence, global citizenship and appreciation of cultural diversity and of culture's contribution to sustainable development" [31]. According to Haubrich [29], the paradigm of SD should be integrated into the teaching of geography at all levels.

In Agenda 21 [32] and in the 2030 Agenda for Sustainable Development [33], the following topics for teaching can be identified: soil/land degradation; desertification; biodiversity/biodiversity loss; climate change; water/oceans; poverty and justice; health and food; consumption; gender differences/gender equality; and housing/safe, resilient, and sustainable human settlements and participation. The former also mentioned environment and development; pollution; agriculture; biotechnology; and new technologies. The latter also emphasize protection of ecosystems, sustainable energy, and sustainable industrialization. Thus, based on these descriptions, it can be stated that suitable teaching topics for education in geography include issues relevant to Earth science and human geography.

Creativity and innovation, creative thinking and problem solving, and communication and collaboration skills help students face future challenges [34]. Creativity includes using imagination, pursuing purpose, being original, and judging values [35]. Imagination involves looking at a situation from a different perspective or thinking of alternatives. Pursuing purpose and being original enable

the student to have a set action or intention while conceiving of new ideas. Creative thinking involves trying new possibilities and rejecting those that do not work. An essential skill that also fits with geography is problem-solving. It involves six key steps: observation of the problem; examination of the potential causes of the problem; identification of alternatives to solve the problem; selection of an approach to solve the problem; implementation of the solution; and verification that the problem has been resolved [34]. Communication and collaboration develop understanding and respect for others' perceptions and arguments, thinking and evaluating one's own personal motives, the relationship of common tasks to one's own competencies, setting your own goals, and thinking about problems and phenomena from different perspectives [29]. Developing communication and collaboration skills also develops global awareness [34]. All of these skills are part of the action competence needed for SD, which should be based on an adequate value orientation [29]. However, many research outcomes have shown that knowledge and skills are not sufficient to produce adequate sustainable behaviour [36–39]. Students should have a chance to experience how beautiful nature can be, how precious culture is, how worthwhile communities and societies are, so they will be ready to protect all these parts of the human–earth ecosystem [29].

### 2.4. Teaching and Learning Principles and Methods Promoting Sustainability in Geography Education

There are several principles in geography education that support the learning of sustainability issues in connection with various teaching and learning methods. According to Haubrich [29], group discussions concerning the structure of a problem, the causes of the problem, and the potential responses to solve the problem from an ecological, economical, and societal point of view are appropriate methods for studying ecological issues. In geography education, it is also important to practice how to behave ecologically, socially, economically, or politically. Group work and projects with activities, such as letters to newspapers and exhibitions at public places, are some well-known teaching methods.

According to Catling [40], geography education should be based on children's geographies or geographical knowledge and the active role of children in their own learning. Valuable environmental experiences can arise when studying takes place with appropriate activities in different environments [41]. The interaction between the student and the place gives birth to the meaning of the place. Meaningful learning experiences support the students' sense of place and identity, arousing their attachment to people and the landscape [5]. According to Hutson [42], "place-based education seeks to connect students to local environments through a variety of principles and strategies that increase environmental awareness and connectedness."

### 2.5. Outdoor Learning Promotes Sustainability in Geography Education

In this study, the definition of outdoor learning is based on the findings of Blenkinsop, Telford, and Morse [43]. They generated a list of pedagogical skills, such as, "becoming more outdoor, environmental, place-based, emergent and/or experiential in the teachers' own practices". The list was based on experienced outdoor school teachers' teaching and learning strategies and the skills that can be regarded as the important skills developed by learners when learning outdoors. Experiential learning was identified as the key teaching and learning strategy in their study. Thus, it is important to pay attention to PBE when learning geography outdoors. According to Way [44], participants acquired meaningful learning on their research topic through PBE. It is an interdisciplinary and experiential learning approach which uses the local environment and society [45].

Outdoor learning is based on holistic, experiential learning, and the integration of knowledge and skills across disciplines [46]. So, learning outdoors can be used for all subjects as well as SE [47,48] for supporting personal and social development, even on a primary school level [49,50]. Moreover, learning outdoors can foster individuals' physical well-being, social and emotional well-being, and deeper levels of learning [51]. Learning in the natural environment develops students' environmental sensitivity, fosters their concrete understanding of environmental issues, and engages students actively with ecological issues [52].

Geography, for its part, supports understanding how people intervene in the world's natural and social processes and, in turn, how spaces, places, landscapes, and environments are affected by such interventions [5,53]. The development of identity, community, and environmental responsibility is supported by a sense of space [54]. Consequently, geography education has a central role in connecting the outdoor physical environment, the sense of space and place, and man's actions in it to support SE.

Creativity is a fundamental part of learning for all learners, but especially for young ones. It requires space and time, as well as a degree of freedom and deep immersion in an area [55]. A variety of multi-modal teaching approaches support this kind of creative learning by involving learners in imaginative experiences and by connecting their learning to their lives [56]. Learning about geographical issues, such as climate change, and developing geographical thinking in outdoor environments offers good opportunities for creativity, which can support the problem-solving skills needed for addressing environmental problems. The outdoor environment provides an opportunity for holistic learning, integrative approaches, and building one's own relationship with nature. Outdoor education also promotes students' concentration, learning, and retention better than teaching and learning in a classroom [57]. These basic skills can be used when confronting the central outdoor environmental issues, and, thus, they support effective learning about sustainability.

Children's connections to the landscape, nature, and people in it can also be developed through outdoor learning, as children learn issues about their living environment and develop their place knowledge. Such learning would be possible if children were studying co-operatively and collaboratively [58] through "ground learning" in local phenomena, gaining "lived experiences" [59]. Consequently, through geographical cross-curricular themes, which foster a strong sense of place and space, understanding of sustainability can also be supported.

## 3. Research Aim and Questions

Promoting sustainability should also be at the heart of geography. Teaching and learning methods play an important role in learning. However, comparative and evaluative studies have not been conducted from the perspective of SE. This study fills this gap. The results can be used to develop geography curricula and teaching in teacher and school education.

The research was guided by the following research questions:

RQ1: What are the teaching and learning methods used in geography education for achieving the sustainable development goals (SDGs) in general and higher education?
RQ2: What are the goals and topics in geography education for achieving SDGs?
RQ3: What are the features of the teaching and learning methods currently used in geography education and SE, including outdoor education, for achieving SDGs?

## 4. Materials and Methods

### 4.1. Data Collection

In this qualitative study, the focus was on articles that concerned geography teaching and SD. The data collection method was a modification of the method used by Jeronen et al. [23]. Articles for the analysis were sourced from scientific databases, such as ERIC, Web of Science, and Education Database. All searches were done in English and conducted in October 2018. The search strategy was based on a systematic organization, categorization, and selection of Boolean keywords related to geography education. A word search was conducted in relation to the terms of geography education and instruction, teaching methods, SD, sustainability, environment, and outdoor education. For each scientific database, a hierarchical search strategy was applied, starting from the simplest combination of Boolean forms and then progressing to more complex forms. Additionally, manual examinations of key journals in geographical education were performed.

The initial searches of the scientific databases and the manual examinations of scientific journals came up with over 2000 search results, from which 52 were chosen for reviewing (covering 2005–2017). However, of these 52 articles, only 17 met the following inclusion criteria and were chosen for the analysis. To be more comparable with the earlier study of the research group [60], the present study ended up with similar data collection criteria applied to geography and its natural and social sciences character:

(a)   Scope: national and international research;
(b)   Type of research: empirical research on teaching methods in geography education and SE, including outdoor education;
(c)   Period: 2008–2018;
(d)   Target groups: students in primary schools, secondary schools, and higher education;
(e)   Language: English;
(f)   Quality: academic papers published in peer-reviewed journals.

The rejected articles focused on curriculum, did not include elements of sustainable development, or included more general descriptions of large courses or education programs, not about the teaching or learning methods themselves. The chosen articles were published in 2008–2018 (Table 1) and their focus groups varied from primary school level to student teachers.

**Table 1.** The journals of the analyzed articles and the teaching and learning target groups.

| Journal | Article Number | Level |
|---|---|---|
| Children's Geographies | [61] | S |
| Education Sciences: Theory and Practice | [62] | S |
| Environmental Education Research | [63,64] | H, P |
| Hydrology and Earth Sciences | [65] | S/ST |
| Interactive Learning Environments | [66] | |
| International Research in Geography and Environmental Education | [67–70] | S, ST, ST/P, S |
| Journal of Computer Assisted Learning | [71] | S |
| Journal of Education for Teaching | [72] | ST/S/H |
| Journal of Geography | [73] | ST |
| Journal of Geography in Higher Education | [74] | ST |
| Journal of Teacher Education for Sustainability | [75] | P |
| Review of International Geographical Education Online | [76,77] | H, P |

Notes: P = primary, S = secondary school, H = high school, ST = student teacher education level, e.g., ST/P student teacher education for primary level.

## 4.2. Analyses

The material was analyzed using content analysis methods [78,79]. Firstly, the material regarding the teaching and learning methods (RQ1) and the topics and goals (RQ2) in geography education for achieving SDGs was divided into two sections for the analyses. The first section contained the entire articles and every teaching and learning method mentioned in them, including the background theory. In the second section, consisting of articles in which the teaching and learning methods, level of knowledge, and thinking skills were studied, only those parts of the articles (excluding the theory part and references) which described the teaching and learning methods used and studied were reviewed. The proportions of the teaching and learning methods used were counted. The proportions were counted with the method of using inductively created Boolean operators and search tools to count the occurrences of each method in each article. Each search result of the Boolean operator was manually checked to verify if it corresponded with the method in question, and then it was counted into the sum of the occurrences. Titles, tables, or references were not included in the sum of the occurrences. Some of the Boolean operators did not result in any occurrences, even though the method had been mentioned in the article. Despite the weaknesses of the method, it was found to be the most effective, as corresponding mentions of each method might be as many as over 90 per article.

Thereafter, deductive analyses were carried out (RQ1) for identifying the level of knowledge [80] and thinking skills [81] included in the teaching and learning methods.

To determine different topics and goals of SDE presented in the expressions collected from the reviewed articles, the expressions were classified deductively according to the UN's definitions of the 17 SDGs (RQ2). These subcategories of the 17 SDGs were then classified into four different main categories: social, cultural, economic, and ecological SD (Table 2).

**Table 2.** The 17 subcategories concerning SDGs.

| Subcategory | Main Category |
|---|---|
| Goal 1. No poverty | Social |
| Goal 2. Zero hunger | Social |
| Goal 3. Good health and well-being | Social |
| Goal 4. Quality education | Social |
| Goal 5. Gender equality | Social |
| Goal 6. Clean water and sanitation | Social |
| Goal 7. Affordable and clean energy | Economic |
| Goal 8. Decent work and economic growth | Economic |
| Goal 9. Industry, innovation and infrastructure | Economic |
| Goal 10. Reduced inequalities | Cultural |
| Goal 11. Sustainable cities and communities | Cultural |
| Goal 12. Responsible consumption and production | Environmental |
| Goal 13. Climate action | Environmental |
| Goal 14. Life below water | Environmental |
| Goal 15. Life on land | Environmental |
| Goal 16. Peace, justice and strong institutions | Cultural |
| Goal 17. Partnership for the goals | Cultural |

The material was also analyzed using a qualitative–quantitative mixed method approach [82] to be able to answer RQ3 concerning features of the teaching and learning methods currently used in geography education and SE, including outdoor education for achieving SDGs. Initially, the articles were read thrice and all substantive expressions concerning features of the teaching and learning methods currently used were written down. After the collection of the expressions, they were inductively classified into subcategories. This subcategorization was repeated independently with the created subcategories to reduce the risk of subjectivity. In the rare case of conflicting subcategories, each of the subcategories was reconsidered, and if any were applicable, they were included in the final subcategorization. The final subcategories were then classified into inductively created higher-order main categories, which enabled us to identify features of the currently used teaching methods.

In order to ensure the reliability of the process, one researcher read the article thrice, and thereafter, all three members of the research groups checked the selections.

## 5. Results

*5.1. The Most Often Used Teaching and Learning Methods in Geography Promoting SE: The Supported Thinking Skills*

The results showed that the teaching and learning methods most often used in geography education for achieving the SDGs in different educational levels were different teaching methods used together, followed by interactive learning (both were noted in 16 of 17 articles). Group work and teacher presentations were mentioned in 12 out of 17 articles (Table A1 in Appendix A). Inquiry-based learning and argumentation were mentioned in half of the articles. Experiential learning (seven articles), information and communication technology (ICT), group discussion, co-operative and collaborative learning (all mentioned in six articles) were also supported, as well as outdoor learning and field work, teacher inquiry, and also games, role plays, and debates were all in five articles.

In counting how many times the term for a certain teaching and learning method was mentioned in the 17 articles altogether in their material, methods, results, and discussion sections, we found interactive learning was mentioned 389 times, followed by ICT, which was mentioned 208 times. The following were mentioned over 100 times in this order: experiential learning (198), group work (144), place-based pedagogy (133), and outdoor learning and field work (118). Problem-oriented and problem-based learning (PBL) and argumentation reached nearly 100 mentions, as did inquiry-based learning. The least popular were both art education methods, such as art instruction, drama, story-line, and reading stories, and service-learning approaches.

The levels of knowledge and thinking skills concerning geography teaching and learning that promotes sustainability were also studied (Table A2 in Appendix A). Fact knowledge and concept knowledge were supported in all articles. Method knowledge was supported in 14 articles and metacognitive knowledge less often, in 11 of the articles.

The lower level of thinking skills (remembering, understanding, and application) were introduced in all 17 articles and analysis and synthesis in 16 articles. The highest level of thinking skills, evaluation, was not present in five articles.

## 5.2. The Topics and Goals in Geography Education Concerning SDGs

In 17 articles on topics in geography education concerning SDGs, there were 461 expressions excluding the theory part of the articles and 584 expressions including these parts. The most often written expressions promoting SD in geography education (Figure 1) were directed to environmental (ecological) sustainability (such as climate change, ground water) (42%), followed by social sustainability (such as gender equality, sustainable lifestyle, healthcare) (25%), and economic (19%) and cultural sustainability (such as traditional cultural knowledge, cultural knowledge of place) (14%). In 11 articles, the sustainability issues were examined both at the local and global levels.

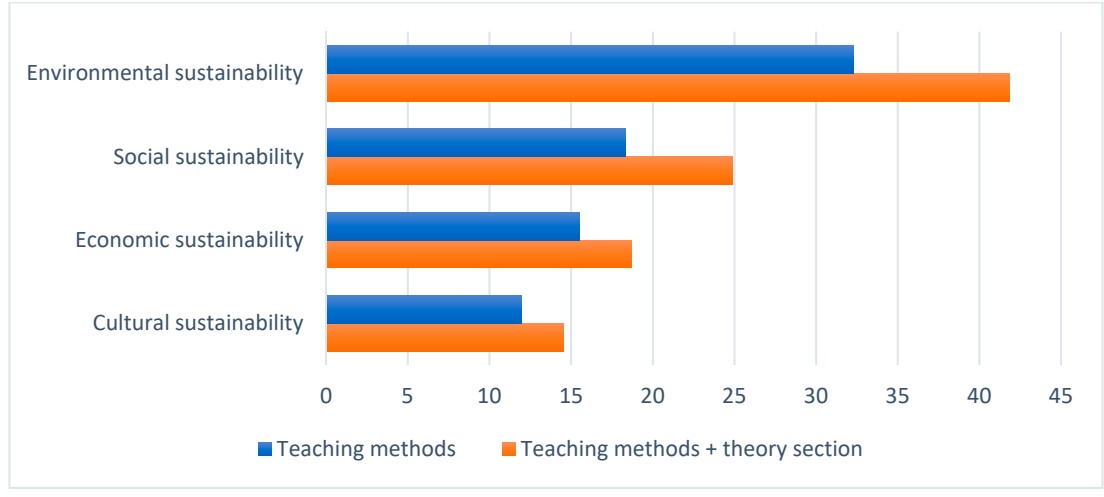

**Figure 1.** Percentage of the dimensions of the sustainability education promoted in geography education, including outdoor education, analyzed in the 17 articles. The four categories were based on expressions in the articles (*n* = 584 expressions including the theory part) and on the stated goals in geography education.

Several goals were identified regarding geography education promoting SE in the 17 analyzed articles: to promote the teaching of SD; to understand the interdependency between humans and nature; to encourage information retrieval and essential discovery; to encourage an interdisciplinary approach; to develop civic skills, system thinking, and sustainable lifestyle; to form your own opinion; to develop pedagogical content knowledge of SD; and to understand the forming and maintaining of ground water, positive future images, and awakening environmental sensitivity.

*5.3. The Features of the Teaching and Learning Methods Currently Used in Geography Education and Sustainable Education, Including Outdoor Education, for Achieving SDGs*

Altogether, there were 1427 expressions concerning the features of the teaching and learning methods currently used in geography and SE, including outdoor education, for achieving SDGs. They were various. One example of these expressions was, "I used diaries as a medium for discussion, holding a 'conversation' with students about their learning of sustainable development across the course of one academic year", which reflected interactive skills between the teacher and students. All in all, 105 different categories of the expressions were classified and, among others, included active participation, thinking skills, animation, evaluation, dialog, demonstrations, and ICT-skills.

General teaching and learning skills (Figure 2) were expressed 636 times as ways to support SD in geography, such as developing skills of lifelong learning (expressed in 19%) and global citizen skills (18% of all expressions). Developing skills for the future was found in only in 2% of the articles about promoting SD in geography. The others were scaffolding, developing interactive skills, developing multi-modal reading skills, and students' self-efficacy and identity.

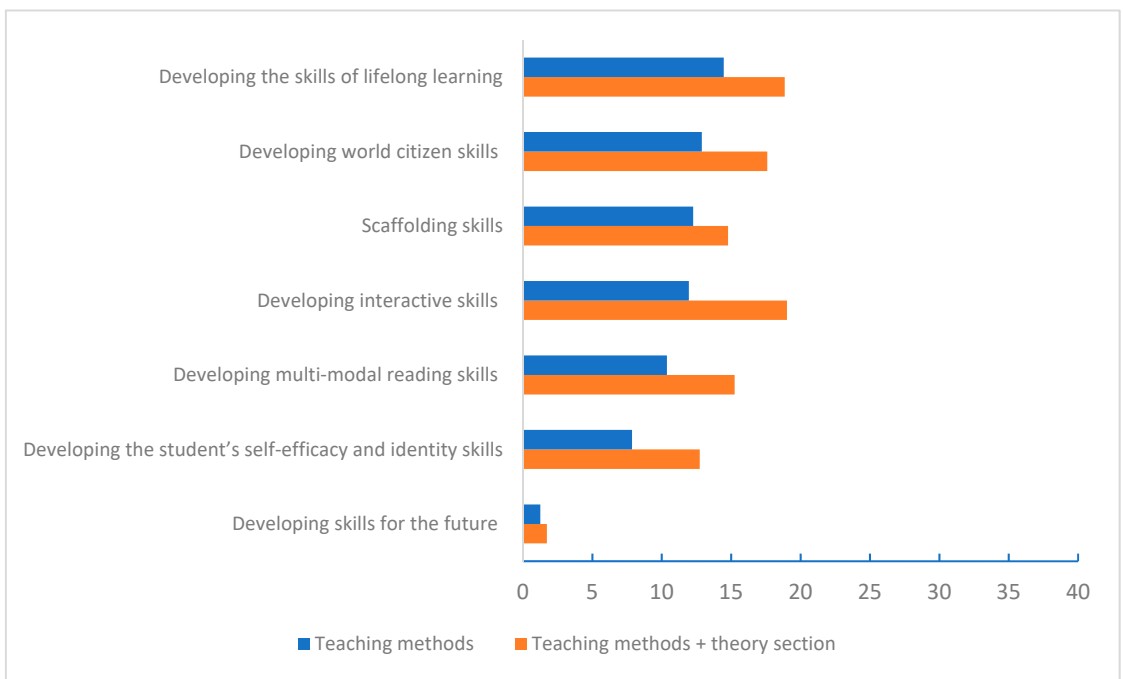

**Figure 2.** Percentage of the general teaching and learning skills in geography that promote sustainable education, including outdoor education, for achieving sustainable development goals (*n* = 636 expressions including the theory part).

Teaching and learning skills in geography (Figure 3) were identified 504 times in the articles about promoting sustainable education, including outdoor education. Among the most often expressed were the skills of scientific research and thinking skills (19%), developing community and co-partnership skills (17%), and place-based learning (17%), and less often mentioned were the general geographical thinking skills and general geographical skills (5%), the use of the senses (4%), and expanded learning environments (<1%).

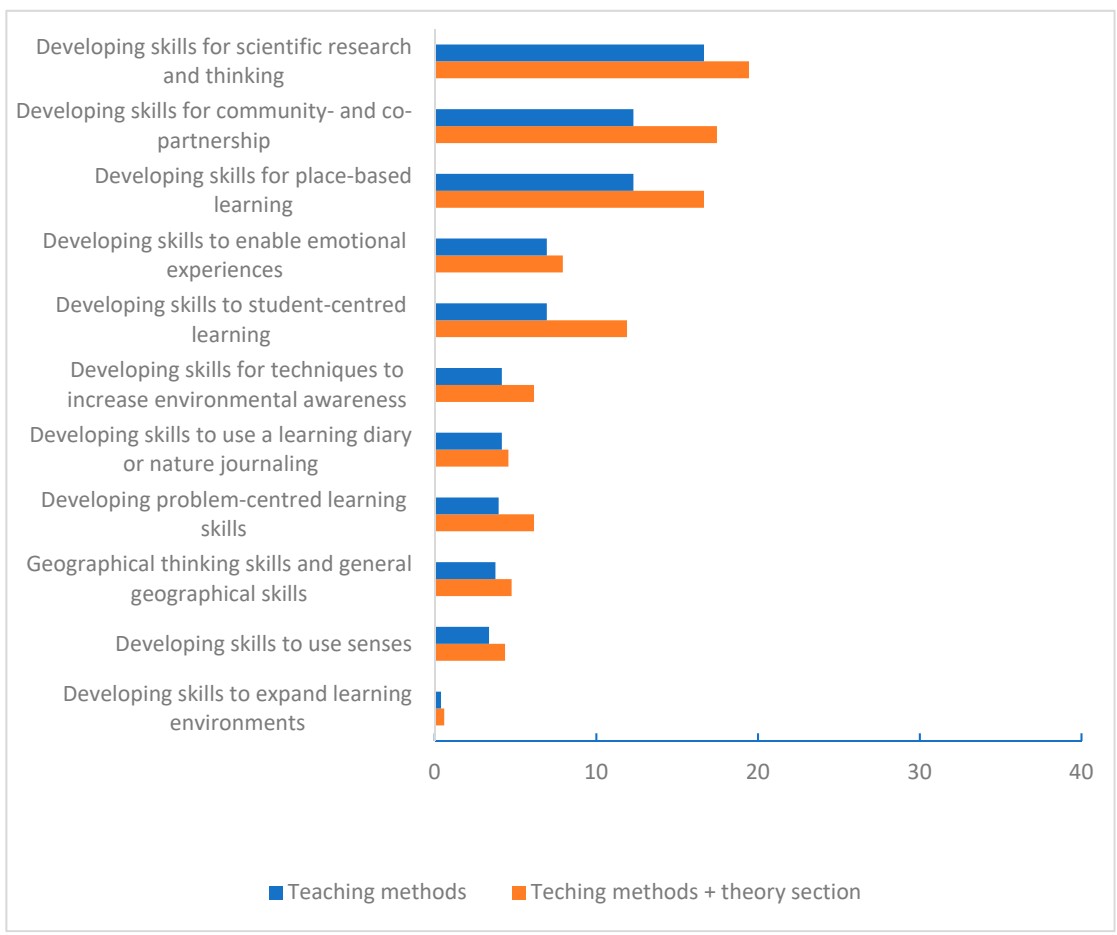

**Figure 3.** Percentage of the expressions (*n* = 504, including the theory part) of teaching and learning skills in geography that promote sustainability, including outdoor education, for achieving sustainable development goals.

Concerning teaching skills in geography, the articles studied mentioned that integrative and interdisciplinary approaches, development of the teaching skills of a teacher, and encouragement of students to be creative are important features to be taken into account.

## 6. Discussion

Citizenship Education and SE support the development of critical thinking skills and help young students understand the complexities of a globalized and interdependent world, as well as reflect on their rights and responsibilities towards each other and the globe. Thus, environmental awareness and consciousness can be seen as important goals, both in geography and in SE [27]. Education in geography fosters knowledge, skills, and concepts for better understanding our own existence and our relationships with other people and the universe [26]. Teaching SD in geography depends not only on high-quality subject matter knowledge, but also on modern pedagogical content knowledge, meaning the teachers' interpretations and transformations of subject matter knowledge support students' learning [14]. Teaching and learning methods should include goal-oriented activities and information exchange between a teacher and students [23]. It is important to study teaching and learning methods, as they have effects on cognitive, affective, and psychomotor learning [83,84].

Yli-Panula et al. [60] emphasized in the earlier study regarding biology teaching methods promoting biodiversity learning that comparative evaluation is needed in relation to the expected results of teaching methods. This study filled the gap concerning geography teaching methods promoting sustainability. The results showed that the teaching and learning methods most often

used in geography education for achieving the SDGs in different educational levels were different teaching methods used together, followed by interactive learning. Using different teaching methods together and incorporating ICT can improve learning. As in the study of biology teaching methods [23], ICT, group work, and teacher presentations were popular. Haubrich [29] recommended group work and discussions as appropriate teaching and learning methods for studying ecological issues. Inquiry-based learning, experiential learning, co-operative and collaborative learning, group discussion, and argumentation were all emphasized, as well as outdoor learning and field work. Co-operation and sharing their own experiences, views, and reasoning in a group helps students understand their own thinking [85] and can support critical thinking and commitment to science education [86]. Through outdoor learning and field work, students' environmental sensitivity, and their concrete understandings of environmental and ecological issues can be developed cf. [52].

The least popular were art education methods, such as art instruction, drama, story-line, and reading stories. The reduced use of these was surprising, as it would be possible to incorporate emotional experiences into studying through these teaching and learning methods, and thus foster individuals' physical, emotional, and social well-being, and deeper levels of learning [51]. Collaborative reading and discussion of read text helps students to understand the text in depth [87]. Although place-based learning was considered important in the reviewed articles, the whole-school approaches were not used at all and service-learning approaches were only mentioned in a few articles. This result is similar to that of a previous study of biology teaching methods by Jeronen et al. [23]. The two approaches mentioned above represent PBL, which enables students to get to know their own neighborhood and community [88,89]. In geography education too, such approaches have been used more extensively than before, as they can generate public interest and motivate local, regional, and national decision-makers to integrate SE into school curricula [90].

The most often mentioned topics promoting SD in geography education were directed to environmental (ecological) sustainability, such as climate change and ground water, followed by social sustainability, such as gender equality, sustainable lifestyle, and healthcare, and then by economic and cultural sustainability, such as traditional cultural knowledge and cultural knowledge of place. The sustainability issues were examined both at local and global levels in most articles.

In this respect, the topics were consistent with the topics presented in Agenda 21 [32] and in the 2030 Agenda for Sustainable Development [33]. There were surprisingly few ecological topics. Issues such as desertification, biodiversity loss, pollution, and protection of ecosystems were hardly addressed. The following economic issues were also missing, consumption, sustainable energy, and sustainable industrialization, which are topics for teaching identified in the lists by the UN [32] and in the 2030 Agenda for Sustainable Development [33].

As important goals in geography education, promoting SD and teaching of SD were mentioned in such a way that students are able to understand the interdependency between humans and nature. Haubrich [29] stressed that individuals should become aware of the impact of their own behaviour. According to UNESCO [31], students should have the possibility to acquire the knowledge and skills needed to understand what human rights, gender equality, a culture of peace and non-violence, global citizenship, and cultural diversity mean. In the articles, it was also emphasized that students should be encouraged to discover information for forming their own opinion. This supports the statement by UNESCO [31], which includes the idea that valuable skills for developing SD are communication and practical and social skills to explore important topics in a range of scales, from local to international. It was also noted as an important goal that teaching should be interdisciplinary, and that it should support and foster positive images of the future, awaken environmental sensitivity, support system thinking, and develop civic skills for sustainable lifestyles cf. [27]. In addition, another important goal was to develop pedagogical content knowledge of SD.

A number of the features of the teaching and learning methods currently used in geography education and SE, including outdoor education, for achieving SDGs features of general learning skills and learning skills in geography were seen to be important. The most important concerning the former

was seen to be the development of lifelong learning and global citizen skills. Surprisingly, "developing skills for the future" was found only in couple of the articles, although it was seen that important goals are fostering positive images of the future and development of civic skills for a sustainable lifestyle to promote SD in geography. The other emphasized features were scaffolding, developing interactive skills, developing multi-modal reading skills, and students' self-efficacy and identity. Among the most often expressed learning skills in geography were the scientific research and thinking skills, developing community and co-partnership skills, and place-based learning, and less often were the geographical thinking skills and general geographical skills and the use of senses. Expanded learning environments were not seen as an important feature of teaching methods, although they could offer good possibilities for PBL [88,89]. Also, Smith [59] argued that geographical education should be based on local phenomena and students' lived experiences. Thus, students, places, and purposeful activities together can produce viable and valuable environmental educational experiences [41], supporting students to develop a sense of place and identity. This again would be a meaningful starting point for a sustained engagement with people and the landscape itself [5].

## 7. Main Conclusions and Implications

This study aimed to identify and describe useful teaching methods in geography education, including outdoor education, to promote sustainability and the goals, topics, and features regarding the teaching methods. Teaching methods in geography which include both physical and human environments cannot be listed from most to least effective, as they are content- and subject-dependent. However, it can be concluded that the whole-school approach, service learning, and future prospect of using different teaching methods together and PBL in learning geography should be implemented in school education to reach the SDGs and to support all four (environmental, social, economic, and cultural) dimensions of SE.

**Author Contributions:** Conceptualization, E.Y.-P., E.J. and P.L.; Formal analysis, E.Y.-P., E.J., P.L.; Investigation, E.Y.-P.; E.J.; Methodology, E.Y.-P.; E.J., P.L.; Supervision, E.Y.-P.; Writing—original draft, E.Y.-P., E.J., P.L.; Writing—review & editing, E.Y.-P., E.J. and P.L. All authors have read and agreed to the published version of the manuscript.

**Funding:** This research received no external funding.

**Conflicts of Interest:** The authors declare no conflicts of interest.

# Appendix A

**Table A1.** The teaching and learning methods mentioned in the 17 articles analyzed.

| Teaching Methods/Article Number | 61 | 62 | 63 | 64 | 65 | 66 | 67 | 68 | 69 | 70 | 71 | 72 | 73 | 74 | 75 | 76 | 77 |
|---|---|---|---|---|---|---|---|---|---|---|---|---|---|---|---|---|---|
| Teacher's presentation | x | | x | x | | x | x | x | | | x | | x | x | | | |
| Teacher's inquiry | x | | x | x | | x | | | | | x | | | | | | |
| Teaching discussion | | | | | | | x | | | | x | | | | | | |
| Group work | | | x | x | | x | x | x | | x | x | x | x | x | x | x | x |
| Co-operative/collaborative learning | | | | x | | | | | | | x | | x | x | | x | x |
| Different teaching methods together | x | | x | x | x | x | x | x | x | x | x | x | x | x | x | x | x |
| ICT | | | | | | x | x | | | | x | | | x | x | x | |
| Hands-on instruction | | | | | | | x | x | | | | x | | | | | x |
| Inquiry-based learning | | | x | x | x | x | x | | x | | | | x | | x | x | |
| Argumentation | | x | x | x | | | x | | | x | x | | | | x | x | |
| Case teaching; Socratic method | | | | x | | | | | | | x | | | x | x | | |
| Interactive learning | x | x | x | x | | x | x | x | x | x | x | x | x | x | x | x | x |
| Problem-oriented/problem-based learning | | | | | | | | | | | x | | | x | | | x |
| Experimental learning | | | | | | | x | | | | | | | | | | |
| Experiential learning | | | | x | | | | x | x | x | | | x | | x | | x |
| Project work | | | | | | | x | | | x | | | x | | x | x | x |
| Service learning | | | | | | | | | | | | | | | | | x |
| Outdoor learning and field work | | | | x | | | | | x | x | | | x | | | | x |
| Place-based pedagogy | | | | x | | | | | | | | | x | | | | x |
| Study trips and visits | | | | | | | | | x | x | | | | | | | x |
| Games, role plays, debates | | | x | | x | | | | x | | | x | | | | | x |
| Art instruction | | | | x | | | | | | | | | | | | | x |
| Drama or story line | | | | x | x | | | | | | | | | | | | |
| Reading stories | | | | x | | | | | | x | | | | | | | |

**Table A2.** The level of thinking skills and knowledge in the 17 articles analyzed.

| Level of Thinking Skills and Knowledge/Article Number | 61 | 62 | 63 | 64 | 65 | 66 | 67 | 68 | 69 | 70 | 71 | 72 | 73 | 74 | 75 | 76 | 77 |
|---|---|---|---|---|---|---|---|---|---|---|---|---|---|---|---|---|---|
| Remembering | x | x | x | x | x | x | x | x | x | x | x | x | x | x | x | x | x |
| Understanding | x | x | x | x | x | x | x | x | x | x | x | x | x | x | x | x | x |
| Application | x | x | x | x | x | x | x | x | x | x | x | x | x | x | x | x | x |
| Analysis | x |  | x | x | x | x | x | x | x | x | x | x | x | x | x | x | x |
| Synthesis | x |  | x | x | x | x | x | x | x | x | x | x | x | x | x | x | x |
| Evaluation | x |  | x | x | x | x |  | x | x | x |  | x | x | x | x |  |  |
| Fact knowledge | x | x | x | x | x | x | x | x | x | x | x | x | x | x | x | x | x |
| Concept knowledge | x | x | x | x | x | x | x | x | x | x | x | x | x | x | x | x | x |
| Method knowledge | x |  | x | x | x |  | x | x | x | x |  | x | x | x | x | x | x |
| Metacognitive knowledge | x |  | x | x |  |  |  | x | x |  |  | x | x | x | x | x | x |

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
