# Peer review of "Teaching and Learning Methods in Geography Promoting Sustainability"

_education, doi:10.3390/educsci10010005_

Round 1

Reviewer 1 Report

The title is relevant and clearly describes the content of the article. Keywords are appropriately chosen by the author. Even if few journals recommend this, an alphabetical order might guide the reader. The introduction presents rather the specialized literature (14 references being included in it) rather than a description of what the author (s) hoped to obtain with accuracy; However, the investigated problem was stated. The literature review was found in point 2 Theoretical background. About half of the article's references are found here (45 references out of 91). The rest are allocated in the part of discussions and even conclusions and implications. A reflection of the majority of references in the part devoted to the specialized literature could give greater consistency to the stage of knowledge. Although the methodology is declared by the authors to be qualitative and there is sufficient information to reproduce a research, the procedures being explained relatively clearly, the figures express numerical data and their name could be more inspired if the explanations of the title were separated or eliminated. The conclusions are relatively supported by the research, the authors feeling the need to strengthen their results with the convictions of other authors through bibliographic references. For the most part, I believe that the article contributes to the development of existing knowledge. The authors could further emphasize their contribution to developing the state of the art in the field, which would increase the chances of the article being cited.

Author Response

The title is relevant and clearly describes the content of the article. Keywords are appropriately chosen by the author. Even if few journals recommend this, an alphabetical order might guide the reader.

First of all, thank you for your comments and time you spent reading and commenting our manuscript.

The key words have been changed to be in alphabetical order.

The introduction presents rather the specialized literature (14 references being included in it) rather than a description of what the author (s) hoped to obtain with accuracy; However, the investigated problem was stated.

It is interesting that the reviewer pointed out the choice we did in writing the introduction. We felt that this topic needs an introduction which leads the readers to the “basic” background content and core concepts instead of description what we hope to obtain. In this way we think to guide the reader to the topic. 

We also wanted to raise up the research gap existing in the literature and of course the aim.

The literature review was found in point 2 Theoretical background. About half of the article's references are found here (45 references out of 91). The rest are allocated in the part of discussions and even conclusions and implications. A reflection of the majority of references in the part devoted to the specialized literature could give greater consistency to the stage of knowledge.

If we understand right the reviewer requests a separate chapter/part should be written devoted to specialized literature to give a consistent view of the stage of knowledge in this field. We had thought about that, but instead of writing this kind of a review article decided to make this a research paper using more like a meta-analytic method and following the writing instructions of the Journal.

Although the methodology is declared by the authors to be qualitative and there is sufficient information to reproduce a research, the procedures being explained relatively clearly, the figures express numerical data and their name could be more inspired if the explanations of the title were separated or eliminated.

If we understand right the reviewer´s comment, we are recommended to delete some information from the figure legends. We have followed the general rules that the figure legends should contain sufficient information for figure message to be understandable separately (from the text), and that is why we don´t want to delete the information in figure legends.

The conclusions are relatively supported by the research, the authors feeling the need to strengthen their results with the convictions of other authors through bibliographic references. For the most part, I believe that the article contributes to the development of existing knowledge. The authors could further emphasize their contribution to developing the state of the art in the field, which would increase the chances of the article being cited.

We have deleted the following sentences. “This was also observed by Wyn et al. [90], who report that the whole-school approach can benefit school communities, which is important especially when teaching sustainability education. The result of the whole-school approach is in line with the review of Jeronen et al. [23] regarding effective biology teaching methods.”

Submission Date

24 November 2019

Date of this review

03 Dec 2019 22:13:30

Reviewer 2 Report

The research problem is interesting and constitutes a contribution to the field of Geography Education, in particular, to knowledge about education for sustainable development. However, at least two points need to be clarified:

The abstract indicates that a qualitative content analysis was applied. However, the study states that mixed methods were used. It is recommended to clarify the methodological positioning of the authors. We remind that the use of frequencies and percentages does not justify the approach of the quantitative methods. It is recommended to justify the reasons why explicitly allude to the biology teaching in a review study on advances in geography teaching. Certainly, both scientific disciplines find points in common. However, this approach may confuse the specific utility of the study in the field of Social Science Education.

Author Response

The research problem is interesting and constitutes a contribution to the field of Geography Education, in particular, to knowledge about education for sustainable development. However, at least two points need to be clarified:

Thank you for your constructive comments.

The abstract indicates that a qualitative content analysis was applied. However, the study states that mixed methods were used. It is recommended to clarify the methodological positioning of the authors. We remind that the use of frequencies and percentages does not justify the approach of the quantitative methods.

The sentence in abstract:

“The data were analysed using qualitative content analysis. A mixed methods approach was used to determine the shares of the teaching and learning methods.”

…has been changed to be

The proportion of teaching and learning methods were determined. The data was analysed using a qualitative content analysis.

,,, in Materials and Methods part:

The proportions of the used teaching and learning methods were counted. The proportions were counted with the method of using inductively created Boolean operators and search tools to count the occurrences of each method in each article.

Some explanation…. We are aware that the definition of mixed method is difficult and there are several interpretations of it. The mixed method can be connected to a mixed data collection, mixed analyses and even mixed research, etc. (see Johnson, B.R., Onwuegbuzie A,J, Turner L.A. 2007. Towards a definition of mixed methods research. Journal of Mixed Methods Research  1(2), 112-133)   We are also aware about the differences concerning qualitative (non-numerical) and quantitative (numerical data) approach, and aware of the connection statistically to the parametric or non-parametric variables and tests. We did not do any statistical tests.

It is recommended to justify the reasons why explicitly allude to the biology teaching in a review study on advances in geography teaching. Certainly, both scientific disciplines find points in common. However, this approach may confuse the specific utility of the study in the field of Social Science Education. 

This sentence: To be more comparable with the earlier study of the research group ´biology teaching methods to promote biodiversity´ [60] the present study ended up to the same criteria. has been changed to be: To be more comparable with the earlier study of the research group [60] the present study ended up to the similar data collection criteria applied to geography and its natural and social sciences character with emphasis on the sustainability aspect. Explanation: The idea to have similar procedure was due to the sustainability (biodiversity part of the sustainability) aspect in both studies. In Finland the same person (nearly in 100%of the cases) teach both geography and biology due to the historical reasons and the position in schools is based on teaching combination of biology and geography  Explanation, to word similar….Similar means in this case that overall form to gather the data was similar. Of course, all the keywords to search were different, etc.

Submission Date

24 November 2019

Date of this review

25 Nov 2019 22:04:51

Round 2

Reviewer 2 Report

Recommendations and suggestions have been addressed.